# A-Scan Parameters and the Risk of Phacomorphic Glaucoma in the Kazakh Population

**DOI:** 10.3390/medicina58111544

**Published:** 2022-10-28

**Authors:** Farida Erkulovna Zhumageldiyeva, Zaure Dzhumataeva, Daniyar Dauletbekov, Marat Suleymenov, Zauresh Utelbayeva, Zhanar Meyermanova, Tynyskul Teleuova

**Affiliations:** 1Department of Ophthalmology, S.D. Asfendiyarov Kazakh National Medical University, Tolebe 94, Almaty 050000, Kazakhstan; 2Department of Ophthalmology, The Central City Clinical Hospital, Zhandosov 6, Almaty 050000, Kazakhstan; 3Department of Functional Diagnostics, Kazakh Research Institute of Eye Diseases, Tolebe 95A, Almaty 050012, Kazakhstan; 4Institute for Ophthalmic Research, University of Tübingen, Elfriede-Aulhorn-Strasse 7D-72076, 72074 Tübingen, Germany

**Keywords:** phacomorphic glaucoma, mature cataract, anterior chamber, axial length, lens

## Abstract

*Background and Objectives:* The aim of our study was to identify risk factors associated with phacomorphic glaucoma (PG) by comparing the biometric parameters of contralateral eyes of patients with PG with the eyes of patients with a mature cataract. *Methods*: This retrospective case–control study included 71 eyes affected with PG, 311 eyes of control participants, and 71 contralateral eyes of patients with PG. All participants were ethnically Kazakh. Axial lengths (AL), anterior chamber depths (ACD), and lens thicknesses (LT) were measured using A-scan ultrasound biometry. To determine the threshold value of the A-scan parameters associated with PG, we performed ROC analysis. *Results*: The eyes with PG had smaller AL and ACD values and larger LT values, followed by the fellow eyes with PG and the control eyes. There were no differences in age and sex between patients with PG and mature cataracts. After adjustment for age and other A-scan parameters, continuous measures of ACD and LT were associated with PG (OR 0.57, 95% CI 0.38–0.73, *p* < 0.001; OR 3.36, 95% CI 1.64–6.912, *p* = 0.001). When A-scan parameters were dichotomized according to the identified threshold, an ACD of less than 2.5 mm (OR 3.113, 95% CI 1.562–6.204, *p* = 0.001) and an LT thicker than 4.75 mm (OR 26.368, 95% CI 9.130–76.158, *p* < 0.001) were found to be related to PG. *Conclusions:* We found that a thicker lens and, possibly, a shallow ACD are risk factors for PG.

## 1. Introduction

Senile cataracts and glaucoma are the leading causes of blindness and low vision worldwide [1,2,3,4]. Glaucoma is a large group of eye disorders characterized by increased intraocular pressure (IOP) that can damage the optic nerve [5]. It is divided into two groups: primary and secondary glaucoma [6]. Phacomorphic glaucoma (PG) is one of the main types of secondary glaucoma, which develops because of a sharp swelling of the lens in a mature cataract [7]. A senile cataract can progress from an immature to a mature state and, sometimes, lead to the development of PG [8]. A mature cataract is a type of cataract in which the lens is opaque (regardless of brunescence), has the complete absence of the red reflex, and is followed by decreased visual acuity [9]. When the cataract matures, the thickness of the lens increases, which creates a closer contact between the iris and the anterior surface of the lens, which leads to the development of a pupillary block [10].

Data on the worldwide distribution of PG are scarce, though it is more common in Asian countries [7,11,12,13,14]. In India, PG is a serious problem, as PG accounts for about 3.91% of the total number of cataract surgeries [11]. PG is frequently associated with a lower socioeconomic class, possibly due to unequal access to ophthalmic screening and surgery [7,11,12]. Wong et al. [4] and Lee et al. [15] *argued that higher incidences of PG in developing countries could be alternatively explained by a lack of education and less near work that leads to shorter* axial length (AL), *which is a* risk for PG. Although PG is currently rare in Western countries, considering increases in the elderly population, the incidence of PG in developed countries is expected to rise [4].

Cataract extraction is the only definitive treatment for PG that is able to reduce IOP in a timely manner before the onset of acute glaucomatous optic neuropathy [11]. In most cases, the prognosis of visual acuity in PG depends on the disease duration and the level of IOP before surgery [12,16,17]. According to Sharanabasamma et al. [12], the prognosis of visual acuity in PG is satisfactory after surgical treatment in eyes with an IOP of up to 35 mmHg and PG symptoms with a duration of less than two weeks. When the IOP is higher than 35 mmHg, optic atrophy occurs early; therefore, it is important to identify biomarkers that can predict the development of PG in patients with a mature cataract to aid early diagnosis.

To date, the main risk factors for PG are an age of older than 60 years, a short AL, a shallower anterior chamber depth (ACD), and female sex [7,8,13]; however, the results of previous studies have been conflicting and inconclusive. For this reason, there is currently no definite eye parameter that can be relied on to predict PG; in addition, there are ethnic differences in eye parameters, which make it difficult to generalize findings across different populations. Generally, studies on this problem have been conducted on small individual samples. Most studies have not noted the role of the initial lens size as a risk factor for PG.

Biometric studies of eyes with PG may provide an explanation for why not all cases with senile cataract lead to PG and, thus, help identify an eye at risk for developing PG. The aim of our study was to identify the parameters associated with PG in patients with a mature cataract by comparing the biometric parameters of the contralateral eyes of patients with PG with the eyes of patients with a mature cataract.

## 2. Materials and Methods

### 2.1. Study Subjects

This retrospective case–control study was approved by the Asfendiyarov Kazakh National Medical University (Almaty, Kazakhstan) and the Central City Clinical Hospital’s (Almaty, Kazakhstan) Human Research Ethics Committee (N: 05-2020, 18 February 2020). This study adhered to the tenets of the Declaration of Helsinki.

This study involved adult patients diagnosed with PG, who had a mature cataract and who were undergoing cataract surgery at the Central City Clinical Hospital at Almaty, Republic of Kazakhstan. The data used in the study comprised the scanned medical records of patients with PG with preoperative measurements from January 2015 to December 2019 and a mature cataract from January to March 2019. During this period, the total number of senile cataract surgeries was 12,008, and PG (233) accounted for about 1.95%. Patients with a mature cataract were chosen as a control group because PG always develops in eyes with a mature cataract; therefore, choosing healthy controls was not appropriate. Furthermore, because PG does not develop in all eyes with a mature cataract, it is important to identify the risk factors for PG in patients with a mature cataract.

The total number of patients with PG identified from the medical records was 233; of these, 141 were ethnically Kazakh, 64 were Caucasian, and 28 were other of Asian ethnicities. Only participants of Kazakh ethnicity were selected. Of 141 Kazakh patients, 70 were excluded for the following reasons: in 41 cases, contralateral eyes with PG were pseudophakic; in 19 cases, the ultrasound measurements of contralateral eyes with PG were missing from the medical records; and, in 10 cases, an acute primary angle closure (APAC) was suspected. The final number of patients with PG included in this study was 71. From all cases with a mature cataract, 313 patients of Kazakh ethnicity with complete medical records were selected.

Inclusion criteria were patients with PG in one eye and a unilateral mature cataract, Kazakh ethnicity, and completed medical records for age, sex, preoperative IOP, ACD between the corneal epithelium, and LT and AL measurements. Patients with other eye diseases, such as uveitis, retinal detachment, primary glaucoma, secondary glaucoma (except for PG), eye trauma, pseudophakic eye, anisometropia, and early ophthalmic surgery, were excluded.

PG was diagnosed when the following criteria were present: an increased IOP above 30 mmHg, conjunctival injection, corneal epithelial edema, shallow anterior chamber, medium dilated pupil, and swollen lens. PG differs from an acute primary angle closure in several characteristics. Patients with PG mainly have a unilateral case, the presence of an intumescent lens in their ill eye (as in the case of an acute attack in the fellow eye), an ACD that is not shallow, and an open anterior chamber angle. In contrast, in acute angle-closure glaucoma, the anterior chamber angle is closed and the fellow eye has a closable angle [7,10,18]. The primary cause of PG is a pupillary block in the thickened opaque lens, and, in acute angle-closure glaucoma, the pupillary block is in pre-existing narrow angles [19].

Here, the IOP was measured using a Maklakov applanation tonometer (model NGM-2, 10 mg, Ocular Instruments Inc., Moscow, Russia). Gonioscopy was not performed due to corneal epithelium edema. The AL, ACD, and LT were measured 10 times using a *10 MHz* A-*scan biometry probe* (A-scan plus, Accutome, Malvern, PA, USA) with applanation after 0.5% proparacaine hydrochloride instillation (Alcaine Alcon-Couvreur, Puurs, Belgium); then, the mean values were estimated. These measurements were taken before cataract surgery and identified from the medical records.

Eyes that met the criteria were classified into three groups: (G1) eyes with PG (71 eyes), (G2) control eyes that had a mature cataract (311 eyes), and (G3) the unaffected eyes of patients with PG (71 eyes). During the phacomorphic attack, the ACD is shallower and the LT is greater than in the pre-phacomorphic state, which is why we cannot rely on these measurements to determine the risk factors for PG. To solve this problem, we compared the contralateral (i.e., unaffected) eyes of patients with PG with the control eyes on the assumption that this would reflect the biometric parameters of the affected eyes before the development of PG because the unaffected eyes of PG patients may represent an early stage of PG.

### 2.2. Statistical Analysis

Descriptive analysis. Data with an asymmetric distribution were analyzed using nonparametric tests. The Mann–Whitney test was performed on age, IOP, AL, examination differences between groups (G1 and G2, G2 and G3), and the results are presented as the median (Me) and interquartile range (IQR). Paired data were analyzed with paired tests. The chi-square test was used in the analysis of categorical variables.

Main analysis. The association between AL, ACD, and LT and the risk of PG was examined using binary logistic regression with adjustments for age, sex, ACD, and LT (G2 and G3). AL and ACD are significantly correlated with each other and ACD is strongly associated with PG according to previous literature; therefore, we included ACD in our multivariate model and excluded AL [8].

ROC analysis was used to identify threshold points for age, AL, ACD, and LT to differentiate eyes with PG and a mature cataract (G1 and G2). The threshold values were determined based on Youden’s index for these variables. In this statistical analysis, the threshold that achieved maximum distance with the chance level was calculated and was referred to as the optimal threshold because it was the threshold that optimized the bioparameters’ differentiating ability when equal weight was accorded to sensitivity and specificity. AL, ACD, and LT were dichotomized using the threshold point based on the Youden index for these variables [20]. Sensitivity, specificity, and OR were calculated for each variable.

Additional analysis. The Spearman correlation was used to identify the relationship of ACD and LT with age. The Mann–Whitney test was used to identify differences in AL between women and men. The statistical significance was defined as *p* < 0.05 for all analyses. Data analysis was performed in SPSS 26.0 (Chicago, IL, USA).

## 3. Results

After excluding eyes with poor image quality and pseudophakic eyes, 71 eyes with PG (age of 69.0 years (63–77)), 71 unaffected eyes of patients with PG, and 311 control eyes with a mature cataract (age of 68.0 years (62–75)) were selected for final analysis. All participants were ethnically Kazakh. The demographic and clinical features of the study subjects are presented in Table 1. There was no significant difference in age and sex between patients with PG and patients with a mature cataract (*p* = 0.197, *p* = 0.182). There was no significant difference in the laterality of disease (right/left eye) between the groups (*p* = 0.144). The IOP was markedly higher in the group with PG than in the mature cataract group (Me: 42 mmHg vs. 14 mmHg, *p* < 0.001). The IOP in the contralateral eyes of patients with PG was 15 mmHg (IQR 14–17).

### Main Analysis

The median values of the A-scan biometry parameters in phacomorphic eyes, contralateral eyes of patients with PG, eyes with a mature cataract, and their association with PG are presented in Table 2. The eyes with PG had the shortest AL followed by the contralateral eyes of patients with PG, and the control groups. Likewise, the eyes with PG had the shallowest ACD and the greatest LT, followed by the contralateral eyes of patients with PG and the control eyes. After adjustment for age, sex, and LT, the continuous measures of ACD and LT were significantly associated with PG (OR 0.57, 95% CI 0.380–0.730, *p* < 0.001 and OR 3.36, 95% CI 1.64–6.912, *p* = 0.001, respectively).

To determine the threshold value associated with PG for the A-scan biometry parameters, we performed ROC analysis. Table 3 summarizes the threshold, sensitivity, and specificity values of A-scan parameters and age, as well as their association with PG. An ACD of less than 2.5 mm and an LT thicker than 4.75 mm were significantly associated with PG (OR 3.313, 95% CI 1.562–6.204, *p* = 0.001 and OR 26.368, 95% CI 9.12–76.158, *p* < 0.001, respectively).

Additional analysis. The results of our study showed that the difference in the median AL between men and women was statistically significant (23.24 mm (IQR 22.77–24.70) vs. (23.07 mm (IQR 22.53–23.80), respectively; *p* = 0.038).

## 4. Discussion

In this case–control study, a shallow ACD and a thicker lens were found to be risk factors associated with PG. Moreover, with advancing age, the depth of the anterior chamber decreased and the thickness of the lens grew, which increased the chances of developing PG in short eyes [21].

Previous studies have shown that elderly age, female sex, a shorter AL, and a shallower ACD are risk factors for developing PG [7,8,10,11,18]. AL is an important parameter for the development of various types of glaucoma. For primary open-angle glaucoma and normal-tension glaucoma, a longer AL is a risk factor [22] and a shorter AL is a risk for primary angle-closure [2]. This conforms to the results of Lee et al. who reported that eyes with an AL shorter than 23.7 mm had a 4.3 times higher risk for developing PG [7]. In contrast, in two other studies undertaken by Subbiah et al. [23] and Mansouri et al. [10], there were no differences in the mean AL between PG eyes and eyes with a mature cataract. Moreover, Keles et al. [8] found that people with an AL < 23.22 mm had a higher risk of PG (OR 1.8, *p* = 0.124), but the result was not significant due to the small number of studied patients with PG. We found a significant difference in the AL between contralateral eyes of patients with PG and eyes with a mature cataract (*p* = 0.001); however, the association between AL and PG diminished after adjustment. The different conclusions of several works are also possibly due to the ethnic differences in the anatomical and biometric parameters of the eye [8,10,15].

Several previous studies did not demonstrate any significant differences in ACD between contralateral eyes with PG and a mature cataract, possibly because of small sample sizes and differences in ethnicity [7,8,23]. In population-based studies on ethnic Chinese in Singapore, the right and left eyes showed a similar pattern of associations; therefore, the unaffected eyes of patients with PG may represent an early stage of PG [24]. In our study, we found a significant difference in ACD between contralateral eyes of patients with PG (G3: 2.48 mm (IQR 2.2–2.6)) and eyes with a mature cataract (G2: 2.7 mm (IQR 2.34–3.1)), *p* < 0.001. The eyes with an ACD ≤ 2.5 mm had a higher risk of PG compared with the eyes with an ACD > 2.5 mm (OR 3.113, 95% CI 1.562–6.204, *p* = 0.001).

Our findings suggest that in addition to the ACD, the LT could also be an important risk factor for PG. The authors of many previous studies did not consider the LT of contralateral eyes of patients with PG as a risk factor for developing PG. For example, Lee et al. [7] and Keles at al. [8] considered that using the contralateral eyes’ LT measurements would be incorrect because these LT values would be different from the pre-phacomorphic state. If the parameters of both eyes are similar, according to the population-based Chinese studies, then we should be able to use the LT of contralateral eyes of patients with PG as a prototype of the pre-phacomorphic lens [24]. In our study, we found a significant difference in the LT between contralateral eyes of patients with PG (G3:4.79 mm (IQR 4.6–4.92)) and the control eyes (G2: 4.56 mm (IQR 4.18–4.78)), *p* < 0.001).

The results of our study suggest that LT should not be ignored as a predictive factor because eyes with an LT ≥ 4.75 mm had a higher risk of PG (OR 26.368, 95% CI 9.130–76.158, *p* < 0.001).

Prophylactic laser iridotomy is not recommended for the contralateral eye of a patient with PG; only lens removal is considered the optimal approach to prevent PG [11]. Therefore, knowing the measures of the LT and considering the AL and ACD parameters, we can assume the risks of PG development in eyes with senile cataracts.

Many studies have confirmed that the risk of developing PG also increases with older age [7,8,10]. In two studies, the average age of patients varied from 72 to 74 years, and the average age of PG patients was significantly higher than that of mature cataract patients [8,10]. In contrast, we found no significant difference in the age of participants with PG (69 years (IQR 63–77)) and a mature cataract (68 years (IQR 62–75, *p* < 0.197)). We hypothesize that Kazakhs have a shorter AL, a smaller ACD, and a thicker lens, which may lead to their accelerated development of PG.

Some authors have noticed that PG is more common among women [7,8,10]. These findings may be related to the fact that women have a shorter AL and a shallower ACD than men [24,25,26]. In our data, there was a statistically significant difference in the AL between men and women (23.24 mm (IQR 22.77–24.70) vs. 23.07 mm (IQR 22.53–23.80), respectively; *p* = 0.038), but there was not any significant difference in sex between the PG and control groups. Another factor that can explain sex differences in the prevalence of PG is longevity, i.e., more women live with PG because women live longer than men [26,27].

This study had some limitations. Firstly, it was a retrospective case–control study. Although every effort was made to obtain patient information, there are still inherent flaws in retrospective studies. Unsfortunately, we were unable to obtain accurate biometric measures of the ACD and LT before the presence of PG because all patients were admitted with an attack of PG. In addition, it would be useful to include information about the anterior chamber angle and refractive status of contralateral eyes. However, the assessment anterior chamber angle of the contralateral eyes and refractive status was not correctly recorded; therefore, these data were considered missing and were not part of the analysis. To determine the exact risk factors for PG, a cohort study would certainly have been better; however, due to the low prevalence of PG in patients with a mature cataract and because it can take decades to develop PG, it is not feasible to design and conduct a cohort study. Secondly, the subjects were of Kazakh origin, and the results of this study may not be generalized to all ethnic groups. Thirdly, in our study, the threshold was taken from the general parameter of the groups under study, but it would be ideal if a population study of the Kazakh ethnicity was conducted to determine the mean eye parameters. In our country, a population-based study of biometric parameters of the Kazakh ethnicity has not been conducted. In the future, if a cross-sectional population study of biometric parameters of the eye between different ethnicities is conducted, perhaps we can obtain answers to our questions about the differences of the biometric parameters of the eye in different ethnicities.

## 5. Conclusions

In conclusion, a thicker lens and a shallow ACD are risk factors for developing PG. Knowing the measures of the ACD and LT parameters, we can anticipate the risks of PG development in eyes with a senile cataract; therefore, these parameters should be included in the screening protocol for older adults with risk factors for PG so that a decision about cataract surgery can be made early, especially in developing countries. Further studies are needed to evaluate the effectiveness of including these parameters in a screening program.

## Figures and Tables

**Table 1 medicina-58-01544-t001:** Demographic and clinical features of the study subjects.

	Phacomorphic Glaucoma (*n* = 71)	Mature Cataract (*n* = 311)	*p*-Value
Age (years) Me (IQR) M ± SD	69.0 (63–77) 69.58 ± 9.107	68.0 (62–75) 67.27 ± 10.479	0.197 *
Sex, *n* (%) (male/female)	24 (33.8%)/ 47(66.2%)	134 (43.1%)/ 177 (56.9%)	0.182 **
Eye laterality, *n* (%) (right/left)	36 (50.70%)/ 35 (49.30%)	164 (52.73%)/ 147 (47.27)	0.144 **
IOP (mmHg), Me (IQR) M ± SD	42 (40–50) 43.61 ± 7.94	14 (13–16) 14.75 ± 3.29	**0.001** *

N, number; Measurements are Me, median; IQR, interquartile range; IOP, intraocular pressure; * Mann–Whitney test; M, mean; SD, standard deviation; ** Chi-square test; Bold value indicates statistically significant results.

**Table 2 medicina-58-01544-t002:** A-scan Parameters in Phacomorphic Eyes, Contralateral Eyes of Phacomorphic Glaucoma, Eyes with Mature Cataract, and Results of Logistic Regression Analysis for Group G2 and G3.

	Group 1: PG (*n* = 71)	Group 2: Mature Cataract (*n* = 311)	Group 3: Contralateral Eyes of PG (*n* = 71)	Crude Odds Ratio (95% CI) * G2 and G3	*p* * G2 and G3	Adjusted Odds Ratio ** (95% CI) G2 and G3	*p* ** G2 and G3
Age (years), Me (IQR) M ± SD	69 (63–77) 69.58 ± 9.107	68 (62–75) 67.27 ± 10.479	-	1.023 (0.997–1.050)	0.088	1.017 (0.99–1.045)	0.213
Sex (M/F)	24 (33.8%)/ 47(66.2%)	134 (43.1%)/ 177 (56.9%)	-	0.674 (0.393–1.158)	0.153	0.73 (0.413–1.293)	0.281
AL (mm), Me (IQR) M ± SD	22.71 (22.12–23.39) 22.82 ± 0.88	23.2 (22.64–23.94) 23.43 ± 1.2	22.79 (22.3–23.4) 22.80 ± 0.73	**-**	**-**	**-**	**-**
ACD (mm), Me (IQR) M ± SD	2.06 (1.9–2.3) 2.17 ± 0.36	2.7 (2.34–3.1) 2.72 ± 0.47	2.48 (2.2–2.6) 2.52 ± 0.43	0.43 (0.18–0.100)	**0.001**	0.57 (0.380–0.730)	**<** **0.001**
LT (mm), Me (IQR) M ± SD	5.03 (4.9–5.22) 5.12 ± 0.36	4.56 (4.18–4.78) 4.47 ± 0.43	4.79 (4.6–4.92) 4.71 ± 0.46	43.817 (17.005–112.905)	**<** **0.001**	3.36 (1.64–6.912)	**0.001**

Measurements are Me, median; IQR, interquartile range; M, mean; SD, standard deviation; M, male; F, female; PG, phacomorphic glaucoma; N, number; G, group; *p* *, crude logistic regression analysis; *p* **, adjusted logistic regression analysis; CI, confidence interval; Bold values indicates statistically significant results.

**Table 3 medicina-58-01544-t003:** The threshold values and results of logistic regression analysis of A-scan parameters.

	Threshold	Sensitivity (%)	Specificity (%)	PG (G1 = 71)	Mature Cataract (G2 = 311)	Crude Odds Ratio (95%CI)	*p* *	Adjusted Odds Ratio (95%CI)	*p* **
				Number of Cases *n* (%)					
AL (mm)	<23.28	69	46	49 (69.01)	159 (51.12)	-	-	-	-
ACD (mm)	<2.5	79	67	56 (78.88)	101 (32.47)	7.762 (4.187–14.390)	**0.001**	3.113 (1.562–6.204)	**0.001**
LT (mm)	≥4.75	94	70	67 (94.37)	92 (29.60)	43.817 (17.005–112.905)	**<** **0.001**	26.368 (9.130–76.158)	**<** **0.001**
Age (years)	≥67.5	55	46	39 (54.93)	168 (54.02)	1.037 (0.618–1.741)	0.89	0.800 (0.428–1.498)	0.800

PG, phacomorphic glaucoma; G, group; N, number; CI, confidence interval; *p*
*****, crude logistic regression analysis; *p* **, adjusted logistic regression analysis; Bold values indicates statistically significant results.

## Data Availability

The data published in this research are available on request from the first author (F.Z.).

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
