# Peer review of "A-Scan Parameters and the Risk of Phacomorphic Glaucoma in the Kazakh Population"

_medicina, 2022, doi:10.3390/medicina58111544_

Round 1

Reviewer 1 Report (Previous Reviewer 2)

All asked suggestions are made.

Author Response

Response Document

We appreciate the time and effort that you and the reviewers have dedicated to providing your valuable feedback on my manuscript.

We look forward to hearing from you again in the near future.

Many thanks in advance.

Farida Zhumageldiyeva

Manuscript number - 1979295

Manuscript Title: A-scan Parameters and the Risk of Phacomorphic Glaucoma in the Kazakh Population

Reviewer reports:

Reviewer 1: All asked suggestions are made.

  1. Response: We appreciate the reviewer’s positive comments.

Reviewer 2 Report (New Reviewer)

No more suggestions required. Mostly all the corrections are covered.

Author Response

Response Document

We appreciate the time and effort that you and the reviewers have dedicated to providing your valuable feedback on my manuscript.

We look forward to hearing from you again in the near future.

Many thanks in advance.

Farida Zhumageldiyeva

Manuscript number - 1979295

Manuscript Title: A-scan Parameters and the Risk of Phacomorphic Glaucoma in the Kazakh Population

Reviewer reports:

Reviewer 2: No more suggestions required. Mostly all the corrections are covered.

  1. Response: We appreciate the reviewer’s positive comments.

Reviewer 3 Report (New Reviewer)

Dear authors,

The manuscript becomes valuable if tables are reconstruction to be clearer for researchers.

with best regards,

Author Response

Response Document

We would like to thank the reviewers and Editor for the constructive comments on our manuscript and for the opportunity to revise and improve our manuscript.

We appreciate the time and effort that you have dedicated to providing your valuable feedback on our manuscript.  We look forward to hearing from you again in the near future.

Many thanks in advance.

Farida Zhumageldiyeva

Manuscript number - 1979295

Manuscript Title: A-scan Parameters and the Risk of Phacomorphic Glaucoma in the Kazakh Population

Reviewer reports:

Reviewer 1: The manuscript becomes valuable if tables are reconstruction to be clearer for researchers.

Response: Thank you. We appreciate the reviewer`s insightful suggestion and we agree with this comment. We reconstructed all tables.

Action: As suggested we reconstructed Table 1, 2, 3.

This manuscript is a resubmission of an earlier submission. The following is a list of the peer review reports and author responses from that submission.

Round 1

Reviewer 1 Report

The paper addresses a well known topic - the high prevalence of angle-closure glaucoma in small eyes with large lenses. The (relative) novelty of the paper is: the cohort of patients ( Kazakh nationality) and the attempt to provide metrics to help identifying patients at risk. 

There are some problems for the second issue: (1) Despite the hard work, sensitivity and specificity are still much too low to state ACD lower than 2.9 mm will induce glaucoma. (2) The higher prevalence in women might be due to older age, please check.

In addition the paper should be thoroughly revised to correct for the frequent inaccuracies.

Author Response

We would like to thank the reviewers and Editor for the constructive comments on our manuscript and for the opportunity to revise and improve our manuscript. We have revised the paper based on these comments and your guidance. For your convenience we have attached a clean copy and a marked copy with all changes highlighted.  The following is the point-by-point answers to address these comments and to summarise the changes we have made in the revision.

We hope that we have addressed the suggestions made by the reviewers and believe that the revisions have improved the quality of the paper.  We look forward to hearing from you again in the near future. 

Many thanks in advance.

Farida Zhumageldiyeva

Reviewer 2 Report

This is an article entitled “A-scan Parameters and Risk of Phacomorphic Glaucoma Among Kazakh Population (medicina-1846950)” which evaluates the risk factors associated with phacomorphic glaucoma in patients with mature cataract by comparing biometric parameters of contralateral eyes of patients.

English needs major revision.

Abstract

-         -  Please write what PG means.

-        -   Please rewrite the first sentence of the abtsract. It is very confusing and complicated.

Introduction

-      -  Ok.

Materials & Methods

-         -  The third paragraph is confusing. Please rewrite to make it clear.

-          - How was the anterior chamber angle of the contralateral eyes of the PG cases? Please admit. As they might actually have narrow angles.

-          - How was the refractive status of the contralateral eyes? Please admit. Were they hyperopic?

-          - Did any of the cases have trauma history? Or did you exclude tarumatic eyes?

Results

-          - Please add the standart deviations of the data.

Discussion

-        -  Please also add a paragraph discussing the refractive status of PG cases.

Conclusion

-         - Ok.

References

-          - Ok.

Tables

-         -  Please add the standart deviations of the data.

Author Response

(The authors gave the same response as above.)

Round 2

Reviewer 2 Report

All asked revisions are made.

Author Response

Response Document

Dear reviewers,
Thank you very much for your consideration, and we really appreciate the comments
and have learned a lot.

Many thanks in advance.

Farida Zhumageldiyeva

Reviewer reports: Comments and Suggestions for Authors: All asked revisions are made. Moderate English changes required.

Response: Thank you very much for your comments. This manuscript was edited for proper
English language, grammar, punctuation, spelling, and overall style by one or more of
the highly qualified native English speaking editors at MDPI English editing. The
editorial certificate will be uploaded with the revised manuscript. I hope the revised
manuscript will meet the requirements of academic publishing in journal Medicina.
